# From sunrise to sunset: Exploring landscape preference through global reactions to ephemeral events captured in georeferenced social media

Alexander Dunkel[1]*, Maximilian C. Hartmann[2], Eva Hauthal[1], Dirk Burghardt[1], Ross S. Purves[2]

1 Institute of Cartography, TU Dresden, Dresden, Germany, 2 Department of Geography, University of Zurich, Zurich, Switzerland

* alexander.dunkel@tu-dresden.de

## Abstract

Events profoundly influence human-environment interactions. Through repetition, some events manifest and amplify collective behavioral traits, which significantly affects landscapes and their use, meaning, and value. However, the majority of research on reaction to events focuses on case studies, based on spatial subsets of data. This makes it difficult to put observations into context and to isolate sources of noise or bias found in data. As a result, inclusion of perceived aesthetic values, for example, in cultural ecosystem services, as a means to protect and develop landscapes, remains problematic. In this work, we focus on human behavior worldwide by exploring global reactions to sunset and sunrise using two datasets collected from Instagram and Flickr. By focusing on the consistency and reproducibility of results across these datasets, our goal is to contribute to the development of more robust methods for identifying landscape preference using geo-social media data, while also exploring motivations for photographing these particular events. Based on a four facet context model, reactions to sunset and sunrise are explored for Where, Who, What, and When. We further compare reactions across different groups, with the aim of quantifying differences in behavior and information spread. Our results suggest that a balanced assessment of landscape preference across different regions and datasets is possible, which strengthens representativity and exploring the How and Why in particular event contexts. The process of analysis is fully documented, allowing transparent replication and adoption to other events or datasets.

## 1. Introduction

Sunset and sunrise are witnessed and appreciated daily by countless people around the world. The timing and visibility of these ubiquitous events varies according to a host of local anthropogenic and natural environmental factors, including weather, pollution, topography

**Data Availability Statement:** All HyperLogLog data used to produce figures and results in this work (see code Supporting information S1–S9) is made available in a public data repository https://doi.org/ 10.25532/OPARA-200.

**Funding:** This work was supported by the German Research Foundation as part of the priority programme 'Volunteered Geographic Information: Interpretation, Visualisation and Social Computing' (VGIscience, SPP 1894) and the Swiss National Science Foundation (Project No 200021E-186389). The funders had no role in study design, data collection and analysis, decision to publish, or preparation of the manuscript.

**Competing interests:** The authors have declared that no competing interests exist.

and the built environment. Sunsets and sunrises are valued in many cultures through ceremonies, storytelling and art, and often linked to landscape appreciation through paintings and photographs (c.f. [1]). In this paper we take advantage of social media manifestations of this phenomena to explore global patterns using millions of photographs and their descriptions.

The rationale for choosing this event is twofold. Firstly, there is a general lack of reproducibility in studies of behavior using social media because samples, population and the phenomena under observations change between studies [2]. This means that methods and results of case studies are difficult to generalize more broadly. For this reason, we chose an event type with a strong temporal and spatial consistency. Sunset and sunrise were among the few events fulfilling these criteria. Secondly, while the use of geo-social media has rapidly increased in recent years, there is a lack of validation of transferability for methods allowing robust identification of value or worth [3]. This issue is commonly known as *results reproducibility*. Improving results reproducibility ideally requires an experiment with two maximally separated datasets and "finding relationships in the same direction and at similar strength" [3 p38] in both. This is difficult to implement with more 'newsworthy' topics. The consistent global and long-term footprint of the sunset and sunrise let us maximize sample size while simultaneously providing a basis for reproducing results using two datasets, albeit not universally representative but independent, collected from Instagram and Flickr.

Importantly, the subject of this study is not simply one of abstract interest. Understanding *what* is collectively valued *where*, by *whom* and *when* is important because an equitable society should take into account diverse ways of perceiving, valuing and appreciating the world [4]. Indeed, as we grapple with the challenge of developing indicators measuring progress towards the Sustainable Development Goals, the importance of capturing this diversity becomes more apparent (un.org/sustainabledevelopment). In natural resource management, planners require information about the spatial distribution and relative importance of collective values related to the landscape, in order to plan and manage scarce resources, for example by identifying regions under pressure, designating protected areas or channeling visitor flows. A common term to refer to these values in natural resource management is 'landscape preference', relating the relative importance and subjective meanings people attribute to landscapes. Landscape preference is expressed through, for example, repetition of activities, personal experiences and meanings attached to places, or the choices and identities that shape behavior [5 p454]. Ilieva & McPhearson emphasize the overarching significance of such information because "human behavior and values [. . .] are affecting, and may even drive, the future of global sustainability" [6 p553]. Concretely this means, to inform decision makers, it is necessary to identify which parts of the landscape are valued and how they are used.

In recent years, landscape preference researchers have turned to georeferenced social media, or more broadly User Generated Content (UGC), for example in the form of Instagram pictures or Twitter posts [7], as a new data source supplementing traditional approaches. However, these data sources are challenging to work with. They are noisy, biased, difficult to completely sample, and often shared through incompletely documented Application Programming Interfaces (APIs). Not surprisingly, Teles da Mota & Pickering [8] summarize that only five out of 48 papers they surveyed conduct analysis at a global scale.

In landscape preference, comparison with global aspects of a phenomenon is important because it allows putting observations into perspective and validating individual, local patterns. A global perspective could for example allow us to study underlying patterns and fundamental biases of the system, or visualization methods, as a whole. By focusing on one, in principle universally observable event captured in two independent datasets, we aim to:

- Contribute a workflow template for consistent visualization of global patterns of landscape preference in popular social media platforms (Flickr and Instagram) with respect to a single phenomenon

- Assess the robustness and applicability of privacy preserving metrics describing variation in a global event

- Interpret the results of a global study of descriptions of sunrise and sunset in social media from the perspective of landscape preference

## 1.1 Related work

Natural resource management, which through planning and policy seeks to contribute to sustainable development goals, applies a broad range of methods to understand landscape preference, including self-directed participant photography, surveys or visual landscape questionnaires [9–11]. However, scaling these approaches beyond individual landscapes remains costly and complex, and is an important limiting factor in practice [12, 13]. One approach to filling this gap is the use of publicly available social media and crowdsourced geodata [14–18].

Locally, analysis of these data are not fundamentally different to contemporary approaches. Chen, Parkins & Sherren [19], for example, study landscape values through a collection of 10,000 geotagged Instagram posts to assess the impact of two proposed hydroelectric dams in Canada. Using manual filtering techniques, qualitative content analysis methods, and context, the authors describe social and cultural landscape values in the proximity of these projects. Similarly, Langemeyer, Calcagni & Baró [20] explore landscape aesthetics in a province in Spain. Their approach quantifies preference in space, using a 500m grid resolution and counting posts (photos). Making results transferable requires standardization, for example by counting unique users only once at each location [20 p544]. Tieskens, van Zanten, Schulp & Verburg [21], take this idea one step further, explicitly using user counts as a primary measure, randomly selecting one photo per user per square kilometer, to reduce the bias from more active users.

Beyond individual local case studies, a number of authors have demonstrated that new data sources are not only comparable to traditional landscape preference research, but also reliable and reproducible. Wood et al. [22] study the frequency of Flickr users per month for national parks and find a strong correlation between officially reported visitation counts. They propose "Photo User Days (PUD)" as a measure, the cumulative number of unique user visits per day. PUD is commonly used as a quantitative proxy for people's choices, behavior and overall valuation (e.g. [17, 23, 24]). As social media and VGI become increasingly popular data sources, the need to understand the source and nature of differences becomes all the more important. Wartmann & Purves [16] in a study comparing free listing, unstructured text and Flickr tags find that differing approaches overlap with respect to some elements of landscape (e.g. the biophysical) but give quite different results in other aspects (e.g. sense of place).

Obviously, further examination of these issues is urgent. Several gaps in research prove to be ongoing barriers. A primary challenge is that cross-platform comparison of patterns and integration of data is complex [25, 26]. Understandably, most of the work cited above uses a spatial filter as the main entry point of analysis, extracting data for a specific area or region. Other dimensions, such as who, what, or when [27], are either ignored or subsumed as distributional measures under the spatial filter (where), preventing assessment of fundamental biases [28 p299]. Often, filtering beyond spatial selection only seeks to exclude blunders, such as wrongly geotagged information (e.g. [20]), or to classify data into relatively broad categories

such as landcover (e.g. [21]). Consequently, the collected data in these studies may cover "every aspect of the environment and all human environmental experience, recollection and imagination" [13 p270]. The result is an exponential increase of 'incidental variables' that must be considered, making it difficult to pinpoint the why and how of patterns [27].

In addition to these core challenges, protecting the privacy of users is increasingly relevant when working with user-generated content [29]. However, in a review of social media studies assessing nature-based tourism, Teles da Mota & Pickering found that "only 12 [out of 48 papers] referred in some way to ethical issues [such as privacy], and mostly briefly" [8 p7]. This is surprising, given that a variety of privacy-preserving methods are well known, for example K-anonymity [30], for estimating and limiting the risk of re-identification in shared datasets, or Differential Privacy [31], providing exact guarantees for privacy-preservation based on carefully calibrated levels of noise added to datasets. These methods particularly focus on privacy-preserving *publishing* of results. According to Malhotra, Kim & Agarwal, any "act of data collection [...] is the starting point of various information privacy concerns" [32 p338]. To reduce the risk of re-identification at data collection time, a promising direction is provided by Probabilistic Data Structures (PDS) [33]. PDS follow the principle of data minimization (ibid.). One such PDS is HyperLogLog (HLL), a data abstraction format for estimating quantities within guaranteed error bounds [34]. A workflow to use HLL with geo-social media was demonstrated by Dunkel et al. [35] studying user frequency of worldwide Flickr posts and quantifying the effects on privacy.

All of the above means that studies regarding global properties of crowdsourced data are rare in landscape research [8]. Among the few exceptions, van Zanten et al. [10] quantified landscape value at a continental-scale, randomly sampling social media data from Panoramio, Flickr, and Instagram, using keywords to filter content for values such as 'aesthetic enjoyment' or 'outdoor recreation'. They emphasize though that the frequency of geotagged photographs strongly correlates with overall frequentation, resulting in overestimates for urban areas and highly frequented, popular sites.

To shift the initial lens of observation from specific locations, we limit this work to a specific sort of events, sunset and sunrise, and thus as a starting point pose a "what" question. This narrow thematic filter allows us to conduct a more focused description and assessment of contextual variables. These events are entirely ephemeral, but have a profound, measurable impact on human-environment perception and interaction. Unlike many other events, an ability to perceive sunset and sunrise is narrowly bound by time, but almost entirely uncoupled from space. In regard to landscape preference, Howard [36] forms a seminal list of 11 'lenses' that may explain differences in preference and perception, from universal preference factors, to nationality, culture, or social status, to more openly defined categories he terms *insideness*, *activity*, or *medium*. Sunset and sunrise appear to be among the few events that could be assigned to the category of 'universal preference factors' in Howard's list. Although we believe care is needed in making claims of universal appreciation, especially from a western perspective [37], sunset and sunrise offer the promise of an indicator for aesthetic values worldwide.

In the context of online sharing on social media, such as Instagram or Flickr, photographs function as evidence for presence in place and time [38]. Individual photographs reflect different memorable experiences and therefore represent different preference contexts. Since mobile phone cameras have become ubiquitous, the barriers to spontaneous photography have decreased and taking a picture of a sunset or sunrise is trivial (Fig 1). Urry linked this behavior to a literary concept, the hermeneutic circle [39 p129] in the context of collectively repeated and mediated behavior. Sunset and sunrise imagery shared online is strongly linked to such behavior [40], and the focus of our analysis. We apply a privacy preserving approach which also greatly improves efficiency, as proposed by Dunkel et al. [35] in a previous scoping study.

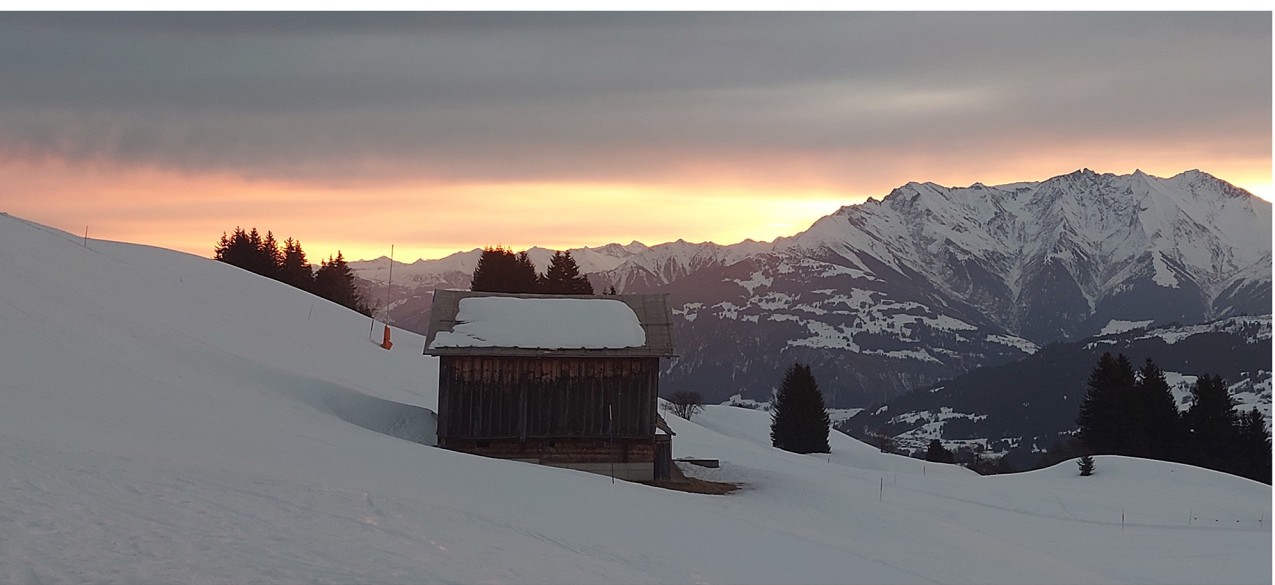

**Fig 1. Sunrise in the Swiss Alps from one of the authors.**

## 2. Materials and methods

### 2.1 Workflow

Our analytical focus was on a workflow utilizing, integrating and improving existing standard methods to explore patterns at a global scale. Our rationale is that using well-known methods allows more attention to be paid to the implementation details, the combination of methods and the analytical process overall. The code used to produce figures in this work is available in Jupyter Notebooks in S1–S9 Files and our workflow is summarized in Fig 2.

### 2.2 Data collection

Data was collected from the public Flickr and Instagram Application Programming Interfaces (APIs) covering periods of 10 years (2007 to 2018) and 5 months (2017/08/01 to 2018/01/04). Our goal was to sample a comparable volume of data for Flickr on the one hand, while reducing incidental variables for Instagram on the other, by covering at least two seasons (Fall and Winter). The lower limit of 2007 for Flickr was chosen based on the year the tagging feature became available. The rationale is that behavior of users and what data could be uploaded is affected by the interface and feature availability (e.g. the tagging field). Therefore, by limiting collection to the time after 2007, we sampled data from a period where the current Flickr feature set was largely consistent and fully developed. Reactions to sunset and sunrise were filtered based on a list of keywords (Table 1) found in the title (Flickr only), the post body, or as explicit hashtags (Instagram) or tags (Flickr). To increase coverage of captured reactions beyond English the keyword list included translations of terms in three European languages: German, Dutch and French. By sampling three additional languages, we aimed to explore the bias which would result if only English was used. Since we also intended to interpret tags and their associations, it was important to use languages which authors of the paper understood, since automatic translations of tags cannot deal with ambiguity and metaphor. Data collection for Instagram was repeated daily using Netlytic [41]. This platform facilitated access to the Instagram API for researchers, but ceased support in December 2018. For the given period, all

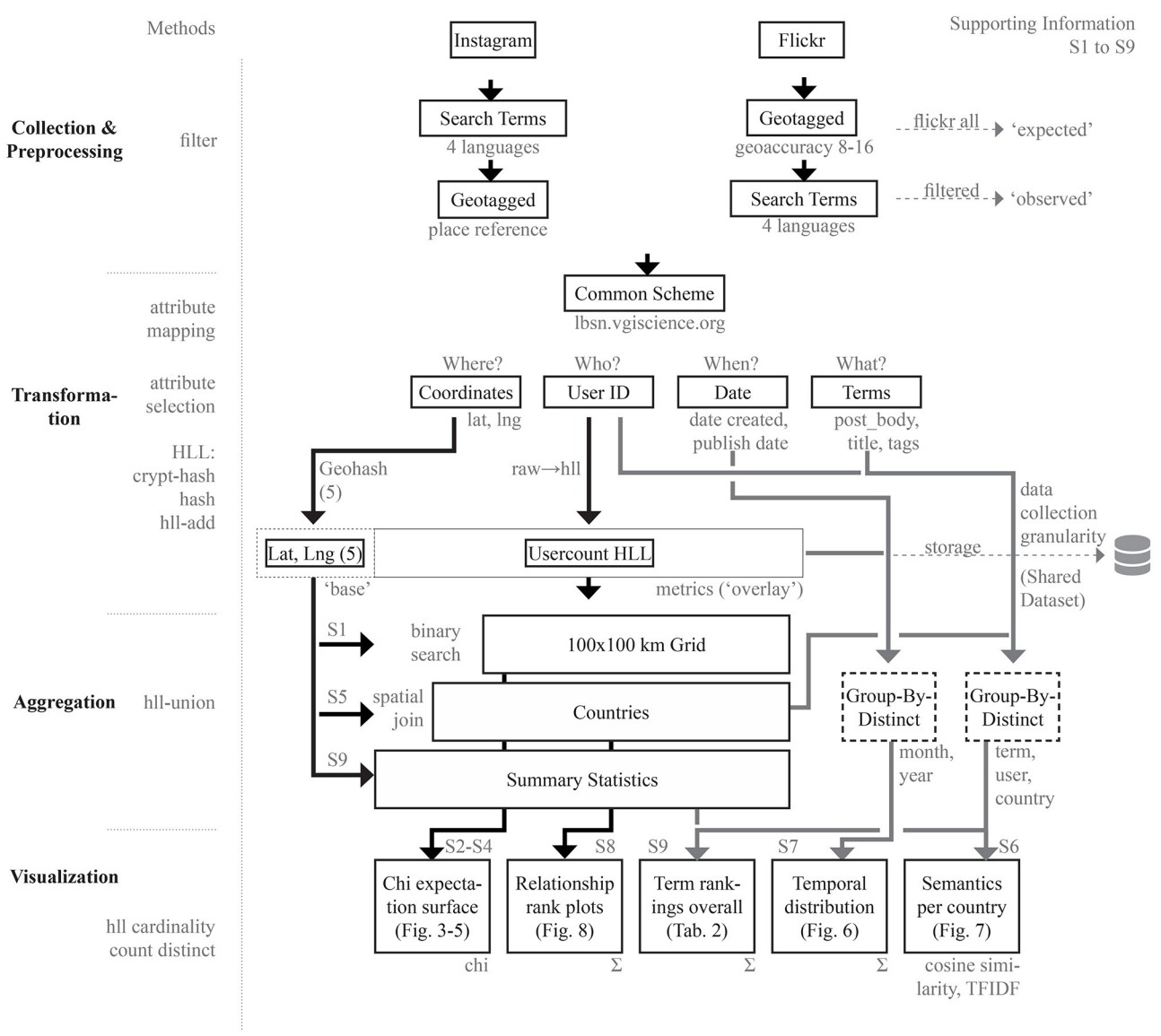

**Fig 2. Workflow schema for data filtering, transformation, aggregation and visualization.**

new posts for each hashtag (Table 1) were retrieved and merged. These hashtag-specific streams include both posts attached to places ('geotagged') and those not explicitly georeferenced. Conversely, Flickr data was retrospectively queried using the API endpoint *flickr.photos.search*, to search for georeferenced posts in the period from 2007 to 2018.

**Table 1. List of keywords, for the four chosen languages and for sunset (top row) and sunrise (bottom row), whose whole occurrence in title, post body, or (hash-) tags defined the initial set of collected reactions.**

| Language ▶ ▼ Event | English | German | Dutch | French |
|---|---|---|---|---|
| Sunset | sunset, sunsets | sonnen-untergang | zonson-dergang | coucherdusoleil, couchersoleil, coucher_soleil, coucher_du_soleil, coucher_de_soleil, coucherdesoleil, coucher AND soleil |
| Sunrise | sunrise, sunrises | sonnen-aufgang | zonsopgang, zonsopkomst | leverdusoleil, leversoleil, lever_du_soleil, lever_soleil, lever_de_soleil, leverdesoleil, lever AND soleil |

This workflow for collecting data leads to a number of sampling effects. Firstly, user groups on Flickr and Instagram are not universally representative, despite having covered together about one billion users in 2017 [42]. Secondly, filters for space, time and language lead to further intra-dataset biases. These are unavoidable when working with these data sources [43]. Our workflow for examining the consistency of results across different groups (Instagram and Flickr), and in terms of separate partitions for "what" is collectively valued, "where", by "whom" and "when" directly reflects this situation. Nonetheless, our results are not fully representative of all users on Flickr and Instagram. Particularly our language filter introduces biases for specific groups using these languages, that we attempt to quantify in §3.3.

## 2.3 Preprocessing

Since different social media platforms have different data structures and attributes, we mapped these to a common structure for comparison [44]. We stored data using a data abstraction format called HyperLogLog (HLL) [34]. HLL estimates the number of distinct items in a set by an irreversible approximation, preventing identification of individual users from collected data and significantly improving data processing performance. At data collection time, the computation of a HLL set requires hashing ids (e.g. a user ID, a post ID), as a means to randomize character distribution [35]. The binary version of hashes is then divided into "buckets" of equal bit length. For each bucket, only the maximum number of leading zeroes is memorized, which frequently means that adding new IDs does not change the HLL signature. Based on the maximum number of zeroes observed for all buckets, it is possible to predict how many distinct items must have been added to a single HLL set, by so-called cardinality estimation.

From a practical viewpoint, the functionality and use of HLL sets is akin to conventional sets. For instance, several HLL sets can be merged (a union operation), to compute the combined count of distinct elements of both sets, without losing accuracy. We use this function to sequentially aggregate data to larger spatial levels in our analysis (§2.5). From a conceptual point of view, HLL effectively summarizes values (in contrast to raw or pseudonymized data), since multiple original IDs can randomly produce the same HLL binary signature. Importantly, we were not interested in individual users, only quantities, and HLL allowed us to reduce the data collection footprint to these quantitative measurements early in the process. Consequently, the study illustrated here can be repeated without the need to store raw data, providing both performance and privacy benefits [35]. Since these additional steps are not relevant to the analytical process and results, they are not described in further detail here. All quantities reported in this paper are estimates, with guaranteed error bounds of ±2.30% [35].

This mode of data collection resulted in two primary datasets. The first contains 3,310,400 Flickr posts for sunset-sunrise reactions between 2007 to 2018 and includes about 2.9 times more sunset (2,545,460) than sunrise (881,320) images. About 3.5% of posts (116,390) contain at least one term from both lists for sunset and sunrise (Table 1). The Instagram dataset, despite covering a much smaller temporal window of only 5 months, contains a larger distinct number of 21,192,990 sunset-sunrise reactions. About 44% (9,462,270) of these posts are geotagged with explicit references to places. With a ratio of 3.7 from sunset (17,660,470) to sunrise (4,741,050) related imagery, this dataset contains a slightly higher proportion of sunset posts. The ratio is consistent with the officially reported number of all posts ever shared on Instagram, for sunset (282,265,750 posts) and sunrise (75,482,380) [45], providing a baseline of validation for the data collection process. About 5.7% (1,208,540) of the Instagram posts contain references to both sunset and sunrise (Table 1).

Two additional datasets were retrieved, with the goal of reflecting a random collection of posts, to be used as the denominator for the chi statistics described in §2.5. For Flickr, the

dataset contains all georeferenced posts (302,101,300) over the same period. For Instagram, the dataset samples a random selection of 20 million posts. Finally, we extracted the subset of references (URLs) to Flickr images with Creative Commons licenses, for qualitative exploration. This included 82,850 images for sunrise and 284,990 images for sunset.

### 2.4 Ethics statement

The use of the datasets was done in compliance to the Flickr, Instagram and Netlytic Terms and Conditions and PLOS ONE requirements for this type of study.

### 2.5 Transformation

In this study, we wished to represent locations, users, time, and semantics, each reflecting one of the four dimensions (Where, Who, When and What) (see Fig 2). The spatial dimension (where) refers to the spatial references and is represented with geographical coordinates for posts. For Instagram coordinates are available for a known gazetteer of 'places' to which images can be related, while in Flickr, coordinates are directly available either as GPS coordinates or through manual geotagging. The temporal dimension (when), represents the time of reaction (taking a photograph). This is directly available on Flickr as the 'post create date', as inferred from the image timestamp. The 'post publish date' is used as a substitute for Instagram because the original date of photo taking is not captured. We use user IDs (who) not to identify individuals, but to allow user counts to be measured. We chose user count as our primary measure since this is not biased by individual prolific posters (Results for photocount and Photo User Days [22] are also available, see S1–S9 Files). Finally, the textual content associated with posts in the form of titles, post body and tags is used to allow analysis of the semantics considered worth reporting by users (what).

### 2.6 Aggregation

This initial data is reduced to a coarser 'data collection granularity' (Fig 2), which is sufficient for worldwide analysis. For coordinates, this means that we 'snap' points to a grid using a Geo-Hash of 5 (see [46]), referring to an average aggregation distance of about four kilometers. Similarly, to explore temporal distributions, dates are grouped to distinct months and years. Distinct terms are selected from the post body (the Flickr photo description and Instagram caption), the post title (Flickr only) and tags (Flickr) or hashtags (Instagram), and used to explore associated semantics (what).

From this initial data collection, measures are stepwise aggregated (1) to a 100x100 km grid, (2) country, and (3) worldwide levels. We chose a 100 km resolution as a balance for the worldwide analysis, after testing with both 50 km and 200 km. Notebooks (S1–S9 Files) allow for exploration of results for arbitrary resolutions and extents. The anonymized data needed to run notebooks and reproduce results are shared in the data repository.

The count of unique elements (i.e. the estimated number of users) are used for visualizing relationships. We chose to use the signed chi value to capture over and under representation of sunset and sunrise, with respect to the overall use of social media, rather than visualizing absolute counts [47–49, p156]. We use a spatial formulation of signed chi values as proposed in an exploratory analysis of social media by Clarke, Wood, Dykes & Slingsby [50]:

$$chi = \frac{((obs * norm) - exp)}{\sqrt{exp}} \quad norm = \frac{\Sigma_{exp}}{\Sigma_{obs}}$$

The resulting chi values take into account the *observed* values (*obs*), as the subset of users taking sunset or sunrise pictures in a spatial unit (e.g. a 100 km grid cell or country), and compare it with a baseline *expected* frequency (*exp*, e.g. the total number of users active in the spatial unit). Clarke et al. [50] refer to this as a "chi expectation surface" (p. 1181). A challenge for calculating chi is that it requires collecting a random sample of social media posts, as a measure for the denominator, the expected frequencies. The simplest approach to generating such a sample is to use the entirety of a social media collection, and indeed for Flickr this approach was possible. However, for Instagram this was not the case. A random selection of 20 million posts was sampled, from a query of the first 50000 posts for each place captured in the core dataset. Since some places have more than 50000 photos, we note that randomness is potentially skewed for Instagram towards highly populated areas or regions. Finally, chi values are only useful where expected counts are sufficiently large, and we used a critical value of chi of 3.84 (1 degree of freedom, $p < 0.05$) to exclude differences which were not significant. This automatically excludes places and regions where not enough photos are available, as indicated with gray color in our figures.

To explore semantic patterns, we used two approaches. We ranked the terms for each country using term-frequency inverse document-frequency (TF-IDF) as a function of their global frequency (inverse document frequency) [51]. We define a 'document' as the set of all terms used by a single user per country. TF-IDF ranks terms used by many users in a country higher than those that are globally common, and ranked lists therefore reveal terms characterizing a grid cell or a country.

$$TF(t,c) = \frac{Number\ of\ users\ using\ term\ t\ in\ country\ d}{Total\ number\ of\ terms\ in\ country}$$

$$IDF(t) = log\ \frac{Total\ number\ of\ countries}{(1 +\ number\ of\ countries\ in\ which\ term\ t\ appears)}$$

$$TFIDF(t,d) = TF(t,c) * IDF(t)$$

To compare semantics between countries, we calculated binary cosine similarity using vectors of all terms used to describe a country (c.f. [16]). Here, binary means that the terms used per country are considered solely based on occurrence, contrary to weighted vectors for terms calculated from the frequency of use. Binary cosine vectors are easier to interpret and allow better comparison of the similarity of vocabularies between countries. Countries reflecting a very similar use of terms tend towards a cosine similarity of 1, while those with very different terms have lower cosine similarities.

$$cosine\ similarity = \frac{\sum_{i=1}^{n} A_i B_i}{\sqrt{\sum_{i=1}^{n} A_i^2}\sqrt{\sum_{i=1}^{n} B_i^2}}$$

where $A_i$ and $B_i$ are binary vectors indicating presence of a term for a particular country.

## 3. Results

### 3.1 Spatial patterns

We first asked whether it is possible to synthesize results in a single map, to compare sunset and sunrise reactions. Fig 3 shows the global pattern of locations where significantly more sunset (red) or sunrise (blue) events were photographed in Flickr than would be expected given

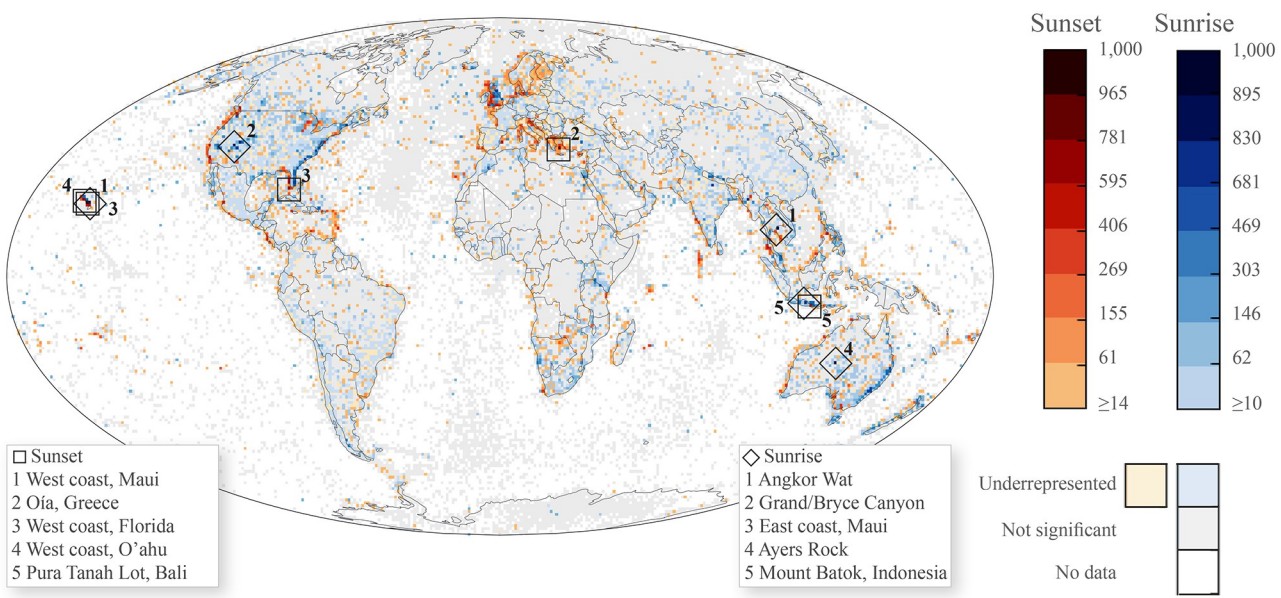

**Fig 3. Chi-value merged—Flickr user count for "sunrise" (blue) and "sunset" (red), 2007–2018.** Focus on positive chi values, normalized to 1–1000 range, Head-Tail-Breaks, 100 km grid. Most significant five grid cells highlighted for sunrise (diamond) and sunset (square).

the underlying distribution of all Flickr images globally. Values are scaled to have the same range (1–1000), and the map therefore illustrates locations where, firstly, sunrise or sunset is often photographed, and secondly, sunrise or sunset events form spatial patterns. The rationale for scaling is that sunset and sunrise are differently valued overall. This makes direct comparison of chi values difficult because more pictures are available on average for sunset (§2.3). Scaling chi values to the same range allows comparison of local event importance, independent of overall valuation. Furthermore, focus is given to positive chi values, which highlights areas featuring over-representation, where either the sunset (red) or the sunrise (blue) attracts significant more attention. An interactive version is available in S4 File. A graphic that shows under- and overrepresentation individually for sunset and sunrise is shown in S1 Fig.

A general trend in Fig 3 is that sunrise events are globally often associated with east coasts (e.g. US, Australia, UK), while sunset events are photographed on west coasts. Southern Europe and many islands appear to be dominated by sunset events, while patterns associated with sunrise also occur in mountainous regions (e.g. the Rockies in the US, Himalayas, and the mountains of Java, Indonesia). As well as these more spatially autocorrelated regions, many individual cells are scattered globally, though with a general tendency away from the global south. Naively, one might expect that these spatial fluctuations are best explained by the *visual qualities* of the sunset and sunrise. That is, locations will exist where particularly picturesque, vibrant or vivid variants of these two events are more often observed. This might, for example, be the case for regions where weather and atmospheric conditions favor stunning sunrises and sunsets. Indeed, the prevalence of coastal locations from where sunset and sunrise can be viewed on a distant horizon, and mountainous locations, speaks to such effects.

But, making sense of the patterns found is more complex. This is most evident when examining extreme values and inspecting sample images found in these cells. For Flickr and sunrise (Fig 3), the highest chi values are in grid cells related to three themes: religious landmarks, famous rock formations and eastern facing coasts. The highest chi value (1000, as scaled from 426) is found at the cell hosting Angkor Wat, a religious temple structure in Cambodia. Cells

with similarly high chi values are found at the Grand and Bryce Canyons in the USA (876); the east coast of Maui, Hawaii, USA (858); Ayers Rock in Australia (847); and at the volcano Mount Batok with *Gunung Penanjakan*, a crater viewpoint in Indonesia (824). Interestingly, while these extreme values all represent popular areas, they do not include societies most frequented city centers, the common focus of ecosystem services assessment for human well-being [52].

For sunset (Fig 3), peak values can be observed at the west coasts of Maui (1000, as scaled from 283) and O'ahu (717), Hawaii; at a western beach stretch of Bali in Indonesia, with Pura Tanah Lot, a waterside Hindu temple (658), and at several cells along the west coast of Florida (776, 600 and 594). When compared to sunrise, landmarks and individual locations appear to have an equally strong influence over preference on Flickr. The city of Oía (Cyclades) in Greece shows the second highest chi value for sunset perception worldwide (931). According to Wikipedia [53], it "provides excellent views of the sunset over the caldera". More supporting evidence can be found, for instance, in a travel magazine [54], which offers a ranking of top spots to view the sunset, with Oía taking first place. However, the peak chi value worldwide for sunset, at the west coast of Maui, Hawaii (1000), is not included in the ranking [54], indicating a bias of information in the travel magazine's report that would be difficult to disprove otherwise. Since these maps are based on user counts, they emphasize the behavior of all users and reduce possible biases from very active individuals.

The patterns for sunset and sunrise reactions on Instagram (Fig 4) largely follow the general trends observed for Flickr. Particularly extreme values for individual cells in Flickr also reflect high significance based on the Instagram dataset. Individual differences in ranking are not surprising, given the fact that Instagram data has been collected for only five months (August to December), compared to Flickr reflecting patterns across all months and from a 10-year time-span. For instance, Oía in Greece is ranked highest for sunset on Instagram, but superseded by the west coast of Maui for Flickr. Maui is known as a popular northern winter travel destination—a season only partially covered in our Instagram dataset. Other differences are more striking, such as the Burning Man festival in Nevada ranking second worldwide for sunrise

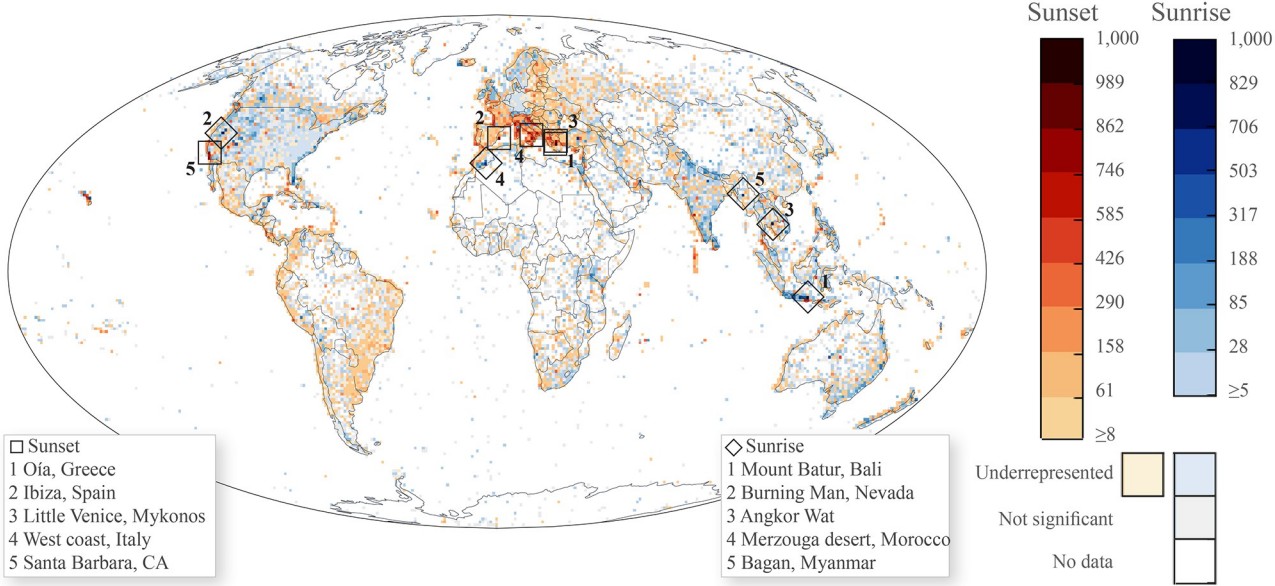

**Fig 4. Chi-value merged—Instagram, user count for "sunrise" (blue) and "sunset" (red), Aug-Dec 2017.** Focus on positive chi values, normalized to 1–1000 range, Head-Tail-Breaks, 100 km grid. Most significant five grid cells highlighted for sunrise (diamond) and sunset (square).

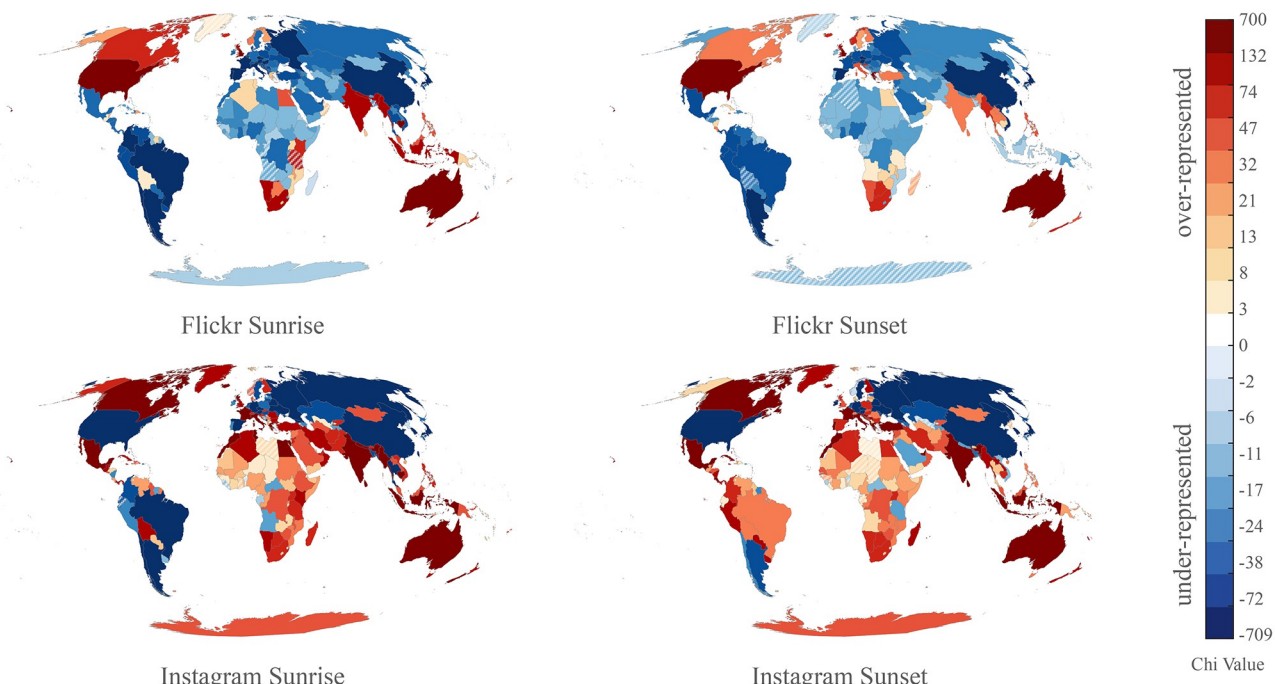

**Fig 5. Chi expectation surface for Countries for Flickr (top, 2007–2018) and Instagram (bottom, Aug-Dec 2017) and sunrise (left) and sunset (right).** Based on user counts, over- and underrepresentation, pooled quantiles classification applied to all four maps. Countries with non-significant results are shown with hatching.

reactions on Instagram. 1295 (±30) users shared sunrise images during the short period of the 2017 Burning Man festival on Instagram, compared to only 54 (±2) Flickr users for the same location over a 10-year timespan.

To explore the patterns at a higher aggregation level, we calculated chi values for countries (Fig 5, code in S5 File). Given the increasing spatial aggregation and reduced influence from individual dominant place characteristics, the differences between Instagram and Flickr are amplified. A general tendency can be observed with Instagram dominating over Flickr for both sunrise and sunset in (e.g.) Southern Europe, Africa or Indonesia, which are known to be tourist destinations. A possible interpretation could be that sunset and sunrise viewing for the Instagram community overall is more prominently favored during travel, whereas preference for sunrise and sunset photography for Flickr users is also strongly linked to exploration of everyday landscapes and local areas, from the user perspective. When focusing on the countries where Flickr and Instagram trends confirm each other, single dominant causal factors become easier to identify. For instance, a popular trend for "guided" sunrise tours [40] for tourists in Indonesia likely leads to positive chi values for both Flickr and Instagram. In contrast, a different positive (Instagram) and negative (Flickr) chi for the sunset suggests that the viewing of this event in this country is affected by a higher variety of 'incidental variables', depending on (e.g.) platform incentives and preferences of certain groups. In other words, while the "sunset" appears to be a significant attraction factor for Instagram users visiting Indonesia, Flickr users' photo behavior is affected by a larger variety of other aspects not captured in our analysis.

### 3.2 Temporal patterns

In addition to the spatial dimension, taking into account temporal and semantic patterns can provide important contextual information supporting our ability to interpret patterns and

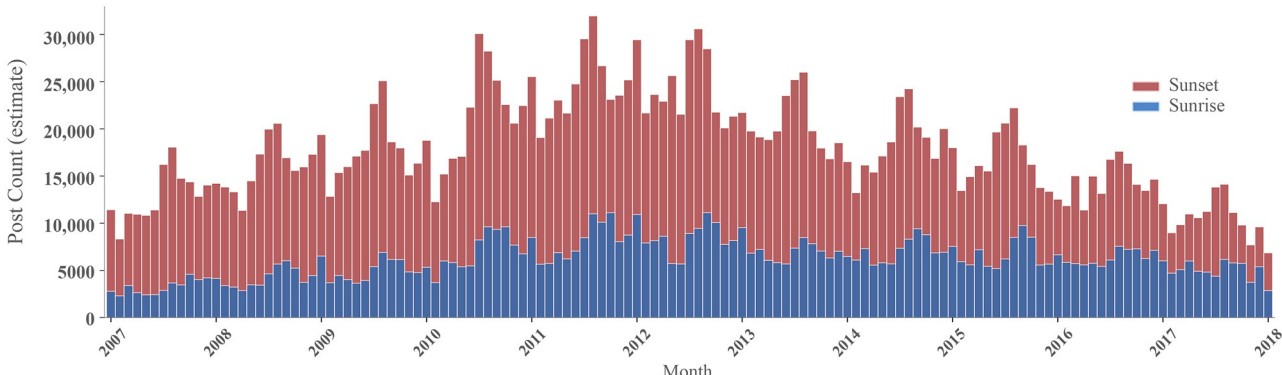

**Fig 6. Temporal distribution of collected data for sunset and sunrise from Flickr.**

draw accurate conclusions. To illustrate temporal relationships, monthly statistics for post count are summarized in Fig 6 for Flickr (see code in S7 File). Since Instagram data were only available for five months, we do not plot a temporal series here.

Several observations can be made. Firstly, the overall proportion of sunrise to sunset images is relatively stable over time, which we interpret as an indication of the reliability of the collected data. Secondly, an increasing trend is visible up to 2012, with a continual decrease afterwards. Overall platform popularity is one explanation [55], with a possible peak in 2012–2013. Finally, despite having collected data worldwide, seasonal trends appear to affect the quantity of shared photos for Flickr. Frequent data peaks can be observed during the northern hemisphere summer season (June, July). Conversely, lows are common during winter months (December, January). These trends may exist due to the bias of the underlying group of users that produced this data. Exploring spatial results (§3.1), it becomes obvious that urban areas and tourist hot spots in Europe and North America are dominantly visible, whereas Africa, South America and large parts of Asia have less data. This is not only a consequence of the language filter that was used (see §2.2). Access to these platforms is limited by origin, income levels, technological knowledge and other factors, generally subsumed under the 'digital divide' [56]. Consequently, the northern hemisphere summer vacation period may reinforce the observed seasonal patterns in data, even for data collected outside Europe or North America due to the effects of tourism.

## 3.3 Semantic patterns

To assess the use of language amongst different users and groups we first explored the use of terms across languages in Flickr and Instagram (Table 2). The ranking is very similar, indicating a universal use of terms to describe sunset and sunrise not significantly influenced by platform. Furthermore, the two top scoring terms "sunset" and "sunrise" are used in 96.7% (Instagram) and 96.9% (Flickr) of all posts in our dataset, revealing a strong preference for English as mode of communication. Thus, if we had searched using only the terms 'sunset' and 'sunrise', we would have been able to capture almost all posts of the other three languages as well. The inverse was also true—of those using language specific terms not in English, 67% of all posts with German, Dutch or French terms *also* contained at least one of the English references (sunrise, sunset, or sunrises or sunsets). We attribute this result to incentives for reaching the broadest possible audience in social media [57].

To explore context with respect to our data, we took advantage of the tags associated with pictures. By calculating TF-IDF values at the country level, we can gain a better understanding

**Table 2. Search terms and comparison of ranking of post count quantities overall for Instagram and Flickr (corresponding code in S9 File).**

| Instagram | | Rank diff | Flickr | |
|---|---|---|---|---|
| **term** | **post count** | | **term** | **post count** |
| sunset | 16,992,173 | 0 | sunset | 2,431,495 |
| sunrise | 4,662,611 | 0 | sunrise | 851,468 |
| sunsets | 1,443,750 | 1 | sonnenuntergang | 112,093 |
| sonnenuntergang | 351,388 | -1 | sunsets | 70,104 |
| sunrises | 126,114 | 4 | sonnenaufgang | 41,513 |
| sonnenaufgang | 103,459 | -1 | zonsondergang | 25,306 |
| zonsondergang | 27,318 | -1 | coucher soleil | 19,603 |
| coucherdusoleil | 12,977 | 2 | leverdesoleil | 10,283 |
| leverdesoleil | 11,494 | -1 | sunrises | 9,627 |
| zonsopkomst | 6,769 | 2 | coucherdusoleil | 7,035 |
| leverdusoleil | 6,080 | 3 | lever soleil | 6,750 |
| zonsopgang | 3,423 | 1 | zonsopkomst | 5,163 |
| lever soleil | 3,386 | -2 | zonsopgang | 3,527 |
| coucher_du_soleil | 2,556 | 2 | leverdusoleil | 2,049 |
| couchersoleil | 2,556 | 0 | couchersoleil | 1,110 |
| coucher soleil | 801 | -9 | coucher_du_soleil | 1,078 |
| leversoleil | 525 | 0 | leversoleil | 186 |
| coucher_soleil | 90 | 0 | coucher_soleil | 31 |
| lever_du_soleil | 72 | 0 | lever_du_soleil | 0 |
| lever_de_soleil | 67 | 0 | lever_de_soleil | 0 |
| lever_soleil | 52 | 0 | lever_soleil | 0 |

of the concepts associated with sunrise or sunset. Fig 7 illustrates TF-IDF values for Zambia and Spain. A few points are evident when we explore the term list for Zambia. First, all of the terms are in English. Second, we find a mixture of generic landscape terms (e.g. river, sky, water, clouds) suggesting elements often photographed in conjunction with sunsets in this context. The prominence of water is unsurprising, given the general preference for water as an element of landscape. The few placenames mentioned (zambezi, african, zimbabwe, victoria-falls) seem likely to refer to the Victoria Falls on the border between Zambia and Zimbabwe, an iconic site likely being photographed by tourists. Other terms also suggest these terms are related to tourist activities at this and other locations (cruise). Although all of the terms associated with Zambia were English, this need not be the case, as is illustrated by the terms retrieved for Spain, where a mixture of English and Spanish is prominent, and the second highest rated term is the Spanish for sunset (which was not a search term). It is also possible to explore similarities between countries, through the words used to describe sunset or sunrise, using cosine similarity. For Zambia we see a strip of neighboring east and south African countries with high cosine similarity, suggesting that the context associated with sunset is more similar in nearby locations. An interactive map, allowing exploration of all countries and the corresponding code is found in S6 File.

## 3.4 Relationships

As observed earlier, a strong consistency of rank order exists across our datasets. In other words, grid cells ranking relatively higher compared to others in Flickr are also likely to show the same relationship for Instagram (Figs 3 and 4). As a last synthesis of information, we therefore generate four relationship plots (Fig 8, see code in S8 File), based on absolute sunrise/

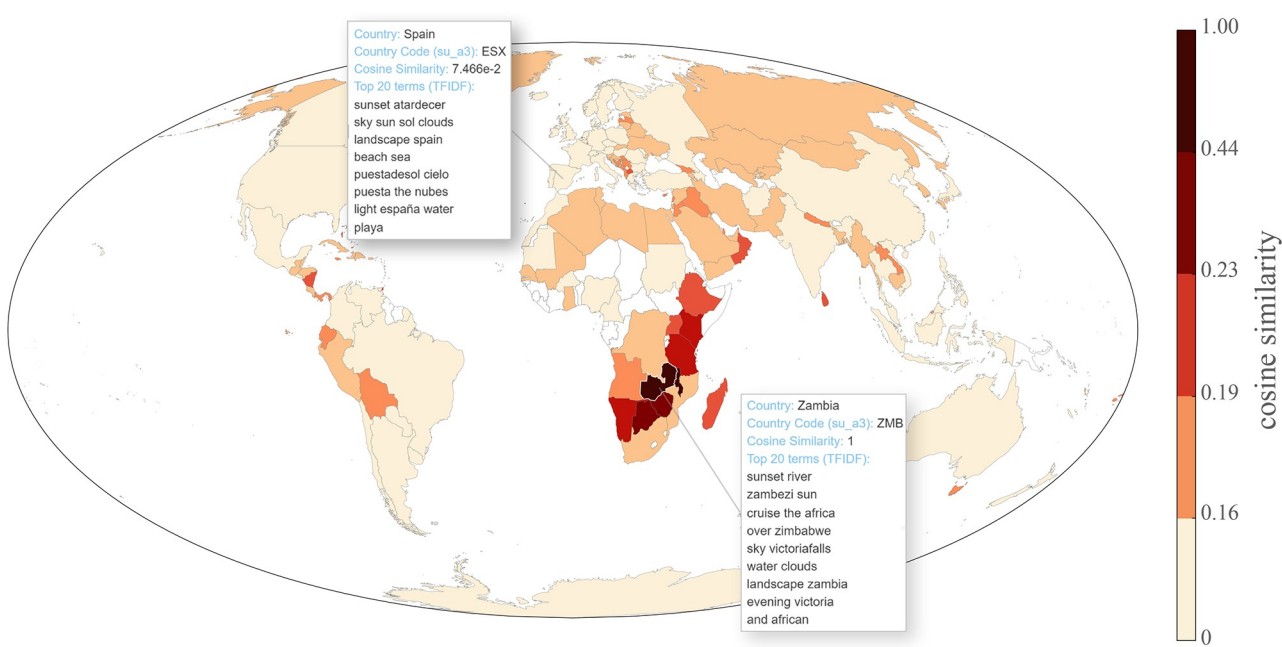

**Fig 7. TF-IDF for top scoring sunset terms in Zambia and Spain and global country similarity ranking based on cosine similarity to Zambia.**

sunset user count ranks as a function of countries and platforms. These plots allow us to explore, firstly, in which countries sunrise and sunset are photographed more often by unique users, secondly, the relationship between the frequency of photographing sunrise and sunset and thirdly, the extent to which behavior depends on the platform.

Generally (Fig 8) we find a strong relationship between particular countries (in Europe and North America) and rank for both Flickr and Instagram. Overall, the strongest relationship between rank of sunset and sunrise photographs was in Instagram ($\rho = 0.97$). However, both Flickr and Instagram have Spearman rank correlations of greater than $\rho = 0.96$, pointing to a generally strong relationship between how often these events are photographed at the level of individual countries in social media. Comparing the two phenomena between different social

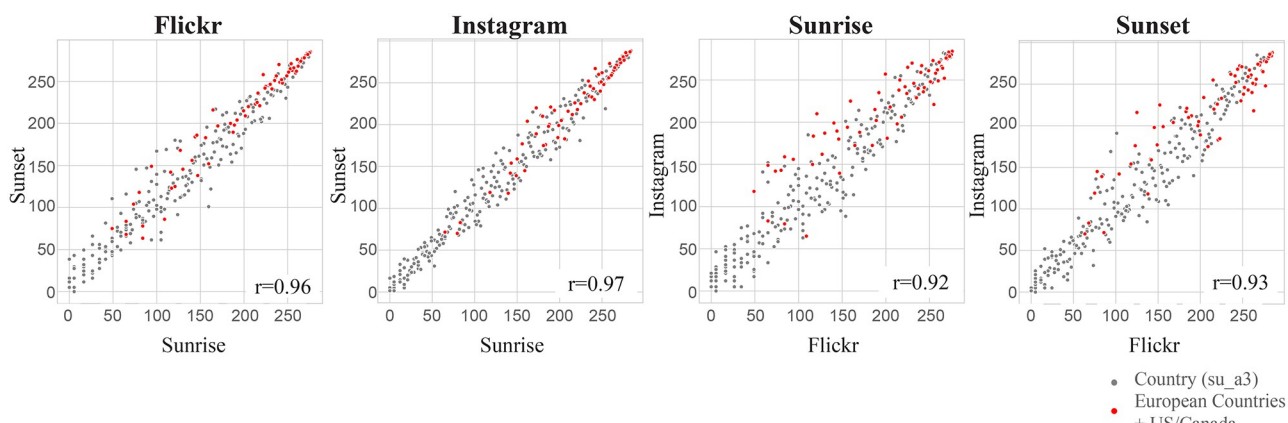

**Fig 8. Relationship between sunset and sunrise (two plots on the left) and Flickr and Instagram (two plots on the right).** Based on ranked (absolute) user count, each dot represents a country (su_a3 codes).

media datasets (Fig 8, two plots on the right), we note that sunset has a slightly stronger correlation between Flickr and Instagram (ρ = 0.93), perhaps suggesting that this event is somewhat more universally appreciated, at least in our data sources, than sunrise. Lastly, European countries and the US/Canada are highlighted in red. These countries tend to dominate the upper ranks for user counts, underpinning earlier observations of western culture dominating on both Instagram and Flickr.

## 4. Discussion

Much research has been conducted around social media for landscape preference studies, but conceptual and practical challenges still hamper application and further development. One challenge is the variability of data and parameter choices complicating comparisons across studies. We explicitly limited the initial set of collected data to a narrow thematic filter—worldwide reactions to sunset and sunrise. This allows us to compare parameter effects in isolation, test the robustness of existing measures and identify opportunities for improvement.

Our results show that it is possible to disconnect the study of landscape preference from overall visitation frequencies. Such a coupling, which appears to be the case in the majority of current studies, may mislead practitioners to overemphasize landscape preference in urban areas and highly frequented popular places, as highlighted by, for example, Teles da Mota & Pickering [8] and Ghermandi & Sinclair [25]. Instead, by using the signed chi square test, we can identify collectively important places and areas (with respect to reactions to sunset and sunrise) at a global scale, independent of overall user frequencies. Furthermore, we also identified places where high user frequencies coincide with an unusually high visual (and virtually communicated) affinity for these events, such as Angkor Wat in Indonesia, Ayers Rock in Australia, or some national parks in the US. For these places and areas, the relative quantification of landscape preference from social media can provide important clues towards their collective use, meaning, and value for social recreation and well-being. Our results are largely consistent across two independent datasets from Flickr and Instagram, indicating a strong consistency of underlying behavior patterns and preference factors for these two events.

For application of results in decision making processes, a number of conceptual challenges remain. Firstly, chi maps show relations and understanding causal factors means zooming into the data. For example, the city of Oía attracts many photographs of sunset while the crater vantage point *Gunung Penanjakan* in Indonesia (§3.1) is a popular location for photographing sunrise. How much of this popularity can be attributed to the global spread of information? Communication, tourist reports and magazine entries (etc.) will amplify certain behavior patterns. It is not possible to fully distinguish between the different motivations for visiting places in hindsight. These self-reinforcing tourist trends are commonly referred to with the *hermeneutic circle* [39 p129]. Whether it is valid to consider these trends as 'collective values' or as indicators for 'landscape preference' is an important theoretical question if we are to use social media data in research. In planning, for example, identifying areas under pressure from mass-tourism [58] or so-called 'cybercascades' [59], warranting action, can be seen as an equally valid outcome, as would protecting 'visually' important areas or developing under-used ones.

We consider it also important to report on negative results and do so in S1–S9 Files. For instance, only user counts proved to be a reliable and robust measurement for collective user attention. User days, as popularized by Wood et al. [22], particularly when measured across several years, as for Flickr, are affected by similar disadvantages to those observed for post (or photo) counts. An example is given in S1 File, where the 100 km grid reveals an unusually high frequency of user days for the grid cell of Berlin. Using Flickr's online search, we manually identified a single user who shared more than 50 thousand photos for sunrise, by what

appeared to be a scripted upload of webcam pictures. Such biases towards individual very active users or bots have been identified by multiple authors [6, 15]. These biases cannot occur when we count unique users, where each user is only counted once (per grid cell, or country). A compromise could be to measure the average number of user days per month, or to consider new measurements such as 'user years', taking into account user visits only once per year per location. This may be particularly useful to focus on repeated attachment to places over longer periods, for specifically capturing the local population's preferences. Notwithstanding these future opportunities, it should be emphasized that visualization of absolute frequencies, still used as a measure in many studies, amplifies inherent biases in user generated content and should be avoided.

Despite the fact that both platforms have different user groups, Instagram users´ place-based preference for sunset and sunrise, in essence, resembles that of Flickr. This confirms the general notion that preference in landscape perception is not random. Turning to classical landscape preference research, the environmental psychologist Stephen Kaplan argues that aesthetics is "an expression of some basic and underlying aspect of the human mind" [60 p242] and later concludes that there are some remarkable communalities in perception between different people, "perhaps in part because of our common evolutionary heritage" [60 p242]. To capture these characteristics, unconscious, implicitly and in-situ collected data is particularly useful [57]. It appears possible to assign at least some of the patterns in our data to these underlying human behavioral traits such as a collective preference for watching the sun rising and setting above water. We sound though a note of caution. Our results are not representative of all cultures, as evidenced by the dominant use of English to describe images, irrespective of search terms in other languages. The patterns we find reflect the preferences of a particular group of individuals expressed globally, rather than a universal preference.

Other patterns required us to zoom into more individual preference factors and characteristics of particular places, or peculiarities resulting from data collection through social media. The opportunities provided by these data sources are manifold. Communalities in perception can be spatially quantified and visualized globally, with unprecedented prospects to improve models of visitation and preference. To this effect the expected frequencies for Instagram and Flickr users per 100 km are made available in the data repository (abstracted HLL data) and can be used to transfer the methods presented in this paper to the exploration of other events and topics. On the other hand, the underlying vagueness in human conceptualization and communication of spatial knowledge makes systematic use of this data difficult [61]. Furthermore, comparable data rarely exists, which means that verification of quality is limited to internal consistency checks [62]. From a systematic model point of view, slicing data into four dimensions (Where, When, What, Who; e.g. [27, 63]), for comparative analysis, and comparing results across these slices, greatly improved assessing the fitness of the collected data. Furthermore, the use of abstracted, estimated non-personal data based on HyperLogLog, was a practically viable solution, supporting a shift towards privacy-preserving and ethically-aware data analytics in research on human preferences [64].

Several caveats apply. A primary challenge of the chi equation is the need to calculate expected values from underlying data whose properties are largely unknown, especially given the growing volumes of these data sources and opaque platform interfaces. Furthermore, we only explored one particular set of reactions, to a narrow selection of two events for the sunset and sunrise. Based on these subsets, extrapolation of results to assess overall landscape experience and preference is not possible. Similar to how landscape preference has been selectively studied using 'landscape inventories' [65], a possible future opportunity could be to consider 'event inventories', as a means to specifically capture temporal landscape preference factors from (e.g.) ephemeral events, such as changing wildlife [66], social events or recurring

phenomena and specific light conditions. This will ultimately improve the information basis that is available to capture how the world is perceived, valued and appreciated.

## 5. Conclusions

By explicitly limiting the initial set of collected data to a narrow thematic filter of millions of worldwide reactions to the sunset and sunrise on social media, we explore possible directions for more robust analysis and visualization of landscape preference. Using the signed chi square test, the results primarily illustrate how this important indicator can be studied without being tied to overall user frequencies or platform, offering an opportunity for a more balanced consideration of collective values in both popular, urban and less frequented, rural areas at a global scale.

Our results address an increasing need for reproducible studies focusing on improving integration of existing methods and standardization, rather than developing new methods. Through the isolation of measures, such as user count, user days, or post counts, and the comparison across datasets (Instagram, Flickr) and dimensions (Who, What, When, Where), a number of pitfalls and issues with data were revealed that may easily invalidate similar studies. Here, the illustrated process can be seen as a blueprint, offering a workflow that can be adapted and transferred to other contexts, beyond reactions to the sunset and sunrise. To this effect, the code for data processing and creation of figures is fully provided in several notebooks, which includes important considerations for data processing, for compatibility with ethical norms in human research.

More broadly, opportunities exist in widening the types of online reactions captured and explicitly considering event inventories. Here, reactions to the setting and rising of the sun captured through social media can be seen as one of many possible indicators for assessing the temporality of human behavior, ephemeral values and transient landscape meaning. Such information may help to better plan and manage a collectively beneficial and equitable development of the environment.

## Supporting information

**S1 File.**
(HTML)

**S2 File.**
(HTML)

**S3 File.**
(HTML)

**S4 File.**
(HTML)

**S5 File.**
(HTML)

**S6 File.**
(HTML)

**S7 File.**
(HTML)

**S8 File.**
(HTML)

**S9 File.**
(HTML)

**S1 Fig. Chi expectation surface for Flickr and Instagram and sunrise and sunset per 100 km grid.** Based on user count, over- and underrepresentation, Natural Breaks classification. (TIF)

## Author Contributions

**Conceptualization:** Alexander Dunkel, Eva Hauthal, Ross S. Purves.

**Data curation:** Alexander Dunkel.

**Formal analysis:** Alexander Dunkel, Maximilian C. Hartmann, Ross S. Purves.

**Funding acquisition:** Eva Hauthal, Dirk Burghardt, Ross S. Purves.

**Investigation:** Alexander Dunkel, Maximilian C. Hartmann.

**Methodology:** Alexander Dunkel, Maximilian C. Hartmann, Ross S. Purves.

**Project administration:** Dirk Burghardt.

**Resources:** Dirk Burghardt.

**Software:** Alexander Dunkel, Maximilian C. Hartmann.

**Supervision:** Dirk Burghardt, Ross S. Purves.

**Validation:** Dirk Burghardt, Ross S. Purves.

**Visualization:** Alexander Dunkel.

**Writing – original draft:** Alexander Dunkel, Ross S. Purves.

**Writing – review & editing:** Alexander Dunkel, Maximilian C. Hartmann, Eva Hauthal, Dirk Burghardt, Ross S. Purves.

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
