## [Decision Letter · Decision Letter 0]

25 Aug 2022

PONE-D-22-16755From sunrise to sunset: Exploring landscape preference through global reactions to ephemeral events captured in georeferenced social mediaPLOS ONE

Dear Dr. Dunkel,

Thank you for submitting your manuscript to PLOS ONE. After careful consideration, we feel that it has merit but does not fully meet PLOS ONE’s publication criteria as it currently stands. Therefore, we invite you to submit a revised version of the manuscript that addresses the points raised during the review process.

We look forward to receiving your revised manuscript.

Kind regards,

Jacinto Estima

Academic Editor

PLOS ONE

Journal Requirements:

Additional Editor Comments:

The paper investigates human reactions to the sunset and the sunrise using geo-referenced photos collected from Instagram and Flickr aiming to understand the motivations behind taking and sharing photographies of sunset and sunrise. The topic of the paper is interesting but relatively narrow and likely not particularly interesting to a broad audience.

There are some minor points to be addressed and other more substancial mostly related to methodological decisions and strategy. Please address all the points raised by the reviewers and try to respond to them one-by-one. That will help improving the paper.

Reviewers' comments:

Reviewer's Responses to Questions

**Comments to the Author**

1. Is the manuscript technically sound, and do the data support the conclusions?

Reviewer #1: Partly

Reviewer #2: Yes

2. Has the statistical analysis been performed appropriately and rigorously? 

Reviewer #1: Yes

Reviewer #2: Yes

3. Have the authors made all data underlying the findings in their manuscript fully available?

Reviewer #1: Yes

Reviewer #2: Yes

4. Is the manuscript presented in an intelligible fashion and written in standard English?

Reviewer #1: Yes

Reviewer #2: Yes

5. Review Comments to the Author

Reviewer #1: The authors use Flickr and Instagram data associated to individual users collected to understand the underlying motivations for taking and sharing sunset and sunrise photography. While the narrative is easily followed and there are some nice figures, the paper is generally colloquial. Overall, I think there are a few issues need to be addressed and clarified further.

1. I recommend that the authors improve the introduction section, especially how the work contribute to the current state-of-the-art.

2. I recommend that the authors improve the results and discussion sections. Currently, the results stay on the description where more in-depth explanations are expected.

3. A figure showing the distribution of photos across countries is helpful. This is needed because if certain countries only have a few data or the distribution is highly skewed, the result is not enough to draw conclusion. In addition, it would be better to add some explanations for the selected time period.

4. There is lack of consideration of bias in the sample of Flickr and Instagram users and the used keywords for collecting data. More discussion is needed with respect to the data representatives. Besides, did the authors consider avoiding fake users or bots running by programming, or did the author conduct an outlier analysis? If yes, please clarify.

5. Why 100x100 km grid? Did the authors do some form of sensitivity test for different size of grid? I’s typically do some simple sensitivity testing to check whether the results are sensitive to resolution.

6. Figures are consistently unclear, or due to text size. They are hard or impossible to read.

Reviewer #2: The paper explores human global reactions to the sunset and the sunrise, using two datasets of geo-referenced photos collected from Instagram and Flickr, and aiming to understand the motivations behind taking and sharing sunset and sunrise photography. The authors attempted to analyze reactions across different groups, and in terms of aspects such as "what" is collectively valued "where", by "whom" and "when." To do this, the authors used relatively simple data analysis methods, specifically (a) using the HyperLogLog algorithm to count the number of distinct users sharing photos related to sunset or sunrise taken at different regions (i.e., different countries, or different cells in a global raster with a resolution of 4km) and at different months, (b) using TF-IDF heuristics to select discriminative terms associated to different regions, (c) using a spatial formulation of signed chi values to select regions that over and under represent the concepts of sunset and the sunrise.

Overall, the paper is both sound and clearly written, presenting the results of what I consider to be an interesting analysis. Still, there are no methodological innovations being proposed (and I do have some questions regarding some of the methodological choices), and the topic addressed in the paper is relatively narrow and likely not particularly interesting to a broad audience.

I have some suggestions in terms of aspects that can perhaps be improved in the manuscript, which I list next.

* The paper should perhaps further justify the choice of languages, besides english, that were considered for analysis with basis on a translation of the terms used for data collection (e.g., German, Dutch, and French). Although I do not consider this a requirement for acceptance of the paper, perhaps other popular languages could have been considered as well, including Spanish, Mandarin, Portuguese, Arabic, or Russian.

* The authors can perhaps further justify the use of HyperLogLog, ideally also presenting a brief explanation on the paper. It is not entirely clear how/if the actual source data will be shared by the authors (the paper mentions a public repository, but at this time I could only access the pre-prepared maps and the Python notebooks. Also, I am not sure if sharing the source data is allowed by Flickr/Instagram), but I would argue that conducting the analysis with privacy-preserving methods would not be the main motivation in using HyperLogLog, although perhaps computational efficiency is an important motivation for this.

* The explanation associated to the use of TF-IDF should be improved. The authors mention the use of "spatial TF-IDF", but it is not clear what the "spatial" adaptation actually is, nor what is the difference towards standard TF-IDF. From the descriptions that are provided latter, I guess the authors are aggregating the textual descriptions from each photo and considering each of the spatial regions (countries or cells) as "individual documents", computing the TF and IDF components with basis on this aggregations (and hence they are able to use TF-IDF to get the most "discriminative" terms associated to each spatial region). However, this is not entirely clear, and should be better explained in the paper. The explanations associated to the TF-IDF equation should also be improved, given for instance that "f" is not a variable used in the equation (whereas "df" is).

* For comparing countries, the authors mention the use of "binary cosine similarity." This should also be further explained in the paper (e.g., does "binary" mean that the authors are considering vectors indicating only the presence of particular terms?) and, ideally, also further justified (why not use vectors of TF-IDF weights?).

* Many of the figures also seem to have a relatively low resolution and, for inclusion in a final manuscript, the authors should ideally provide vector versions of the images, instead of PNG files.

6. PLOS authors have the option to publish the peer review history of their article (what does this mean?). If published, this will include your full peer review and any attached files.

Reviewer #1: No

Reviewer #2: **Yes: **Bruno Martins

---

## [Author Response · Author response to Decision Letter 0]

12 Oct 2022

We would like to thank the editor and the reviewers for their suggestions and insightful comments. We have addressed all suggestions in the revised manuscript. In particular, we have substantially revised the introduction to make clearer our contribution, and carefully revised the paper throughout. We have also provided high resolution versions of all figures, and made these available online and interactively.

Below, we provided detailed point by point responses to each reviewer's comments individually.

> Reviewers' comments:

> Reviewer #1:

> The authors use Flickr and Instagram data associated to individual users collected to understand the underlying motivations for taking and sharing sunset and sunrise photography. While the narrative is easily followed and there are some nice figures, the paper is generally colloquial. Overall, I think there are a few issues need to be addressed and clarified further.

Response: Many thanks for these comments. We are pleased that the narrative is easy to follow - we have edited the manuscript to avoid obvious colloquial use of language. However, we don’t intend to deliberately obfuscate, so we have tried to stick with clear, direct sentences without unnecessary use of complex terminology.

> \\1. I recommend that the authors improve the introduction section, especially how the work contribute to the current state-of-the-art.

Response: 

We have carefully revised the introduction, making clearer our motivations for choosing the sunset and sunrise (and not any more 'newsworthy' event) and how this choice is linked to the state of the art. Specifically, there are several characteristics of this type of event that allow us to significantly reduce the number of 'incidental variables' (while maintaining sample volume) when studying consistency and reproducibility of our method. So far, research using geo-social media in general struggles with the issue of 'results reproducibility' because samples are either too small or data is affected by too many superimposed events included in the user generated data. This usually makes it difficult to generalize methods, which is a critical contribution from the point of our article. 

Importantly, we plan to share the "expected" frequencies for Instagram and Flickr as Supporting Information (data repository S10), meaning that these datasets can be used to verify reproducibility for other topics in other studies (albeit at a granular 100 km level suitable for global studies). Variability of data also partly affects our data, given the global data collection footprint and the general noisiness of geo-social media. 

However, our focus on this single event type at the macro scale allowed for both, a maximization of volume, for statistical significance testing, and a minimization (as far as possible) of incidental variables. Later in the text, we more clearly describe the possible effects of the remaining variables outwith our control. We have also removed a number of some material from the introduction that duplicated the state of the art section.

> \\2. I recommend that the authors improve the results and discussion sections. Currently, the results stay on the description where more in-depth explanations are expected.

Response: We think this comment is to some extent a reflection on the introduction. By rewriting the introduction, as detailed above, and making edits to our results and discussion, we think the paper overall is more coherent.

> \\3. A figure showing the distribution of photos across countries is helpful. This is needed because if certain countries only have a few data or the distribution is highly skewed, the result is not enough to draw conclusion.

Response: This is indeed correct, countries with few observations should be excluded from the analysis. While we did this for other figures, we omitted highlighting non-significant results for the country figure. The issue is addressed in the paper’s revised Fig. 5. While there are a number of non-significant (sub-)countries for both Flickr and Instagram, they are mostly limited to very small countries or islands, which means the effect on the map, albeit noticeable, is not big. For instance, for Instagram and sunset, the total area of countries with significant results is about 143 million km², while the non-significant area covers some 2 million km² or about 2% of the total area. Given the significance test, we believe a separate graphic showing absolute numbers is not necessarily needed. 

Nonetheless, we agree that these numbers may indeed be helpful to readers interested in in-depth examination of our data processing and we updated the supplementary jupyter notebook, which now lists these and other absolute numbers for countries in Supporting Information S5 (05_countries.ipynb, see cells 401ff).

> In addition, it would be better to add some explanations for the selected time period.

Response: We agree and added our rationale for limiting the time periods during data collection, p. 9, ll. 178-184: 

“Our goal was to sample a comparable volume of data for Flickr on the one hand, while reducing incidental variables for Instagram on the other, by covering at least two seasons (Fall and Winter). The lower limit of 2007 for Flickr was chosen based on the year the tagging feature became available. The rationale is that behavior of users and what data could be uploaded is affected by the interface and feature availability (e.g. the tagging field). Therefore, by limiting collection to the time after 2007, we sampled data from a period where the current Flickr feature set was largely consistent and fully developed.”

> \\4. There is lack of consideration of bias in the sample of Flickr and Instagram users and the used keywords for collecting data. More discussion is needed with respect to the data representatives.

Response: We agree that bias and representativity is a crucial topic, particularly in the context of our article. We appreciate the chance to devote more attention in §2.2 (Data Collection) to sampling effects from data collection, specifically from our language filter. The following new paragraph was added (p. 10, ll. 200-209; see also changes in ll. 188-192): 

“This workflow for collecting data leads to a number of sampling effects. Firstly, user groups on Flickr and Instagram are not universally representative, despite having covered together about one billion users in 2017 [42]. Secondly, filters for space, time and language lead to further intra-dataset biases. These are unavoidable when working with these data sources [43]. Our workflow for examining the consistency of results across different groups (Instagram and Flickr), and in terms of separate partitions for "what" is collectively valued, "where", by "whom" and "when” directly reflects this situation. Nonetheless, our results are not fully representative of all users on Flickr and Instagram. Particularly our language filter introduces biases for specific groups using these languages, that we attempt to quantify in §3.3.”

While we improved the description of these biases early in the manuscript, we highlight our investigation of consistency of term use and distribution in §3.3, specifically p.21, ll. 423-430:

“[...] the two top scoring terms “sunset” and “sunrise” in English are used in 96.7% (Instagram) and 96.9% (Flickr) of all posts in our dataset, revealing a strong preference for English as mode of communication. Thus, if we had searched using only the terms ‘sunset’ and ‘sunrise’, we would have been able to capture almost all posts of the other three languages as well. The inverse was also true – of those using language specific terms not in English, 67% of all posts with German, Dutch or French terms *also* contained at least one of the English references (sunrise, sunset, or sunrises or sunsets). We attribute this result to incentives for reaching the broadest possible audience in social media [57].”

In other words, we indeed expected a much stronger effect of our language selection on results, and our work shows, at least for these European languages, that sampling only in English would retrieve a very similar dataset. We provide other researchers with the opportunity to use the intersection capabilities of HyperLogLog to examine further relationships within the data that we share in Supporting Information S10, together with the code in S9 (Jupyter Notebook: 09_statistics.ipynb). 

Lastly, we observe a strong consistency of rank order across our datasets (p.22, ll. 450-469). For instance, the ranking of term use between Instagram and Flickr is largely consistent (Table 2), despite the fact that both networks have a quite different user makeup and languages distribution [43], which speaks, from our point of view, for a low or only moderate data bias from language sampling.

> Besides, did the authors consider avoiding fake users or bots running by programming, or did the author conduct an outlier analysis? If yes, please clarify.

Response: Thank you for pointing this out, we agree that bot detection is important (though perhaps less so in Flickr). We did not conduct an outlier analysis because this would have made it necessary to process individual users in our dataset, which we felt was ethically unjustified and not compatible with the HyperLogLog approach that we used. We did describe and discuss a single observation that refers to such fake users or bots, namely "a single user who shared more than 50 thousand photos for sunrise, by what appeared to be a scripted upload of webcam pictures" (p.25, ll. 511-512). The effect of this user was only noticeable when studied using post count or user days, not user counts. The reason here is that sample volume and the 100x100km grid effectively prevents individual users (or bots) from gaining a significant impact on results. The impact of individual users is also detectable from the ratio between these quantitative measurements. For instance, for the grid cell in Berlin (Flickr, Sunrise), we observe only 444 (estimated) distinct users, but 47,582 user days and 49,597 post count, an extreme outlier given the worldwide average ratio used (and visualized) by chi. This is also why we decided to focus on user counts in our study and emphasize this in the discussion.

Lastly, a good question is if a systematic underlying sampling effect from bots/fake users exists that applies to the whole dataset or specific parts of it. We cannot fully answer this question, since we did not study individual users. However, identifying bots is a complex, non-deterministic process (e.g. see You et al. 2012) and any weakness in such a process or algorithm would have likely also introduced its own sampling effects. Instead, we chose to focus on validating the impact of bots/cyborgs by comparing the consistency of results across two independent datasets (Flickr and Instagram). The consistency of results suggests that our dataset (user count) is not significantly affected by bots or fake users.

You, A., Chu, Z., Gianvecchio, S., Wang, H., Member, S., Jajodia, S., & Member, S. (2012). Detecting Automation of Twitter Accounts: Are You a Human, Bot, or Cyborg? 9(6), 811–825. DOI: 10.1109/TDSC.2012.75

> \\5. Why 100x100 km grid? Did the authors do some form of sensitivity test for different size of grid? I’s typically do some simple sensitivity testing to check whether the results are sensitive to resolution.

Response: Yes, we did this. We tested the output for a 50x50 km grid and found it too small for worldwide analysis (the outputs are included in the data repository, S10, for 50 km). We also tested a granularity of 200 km, which on the other hand we found not granular enough for the manner in which we intended to interpret results. 

Different resolutions can be explored using the data in Supporting Information S10. The initial parameters are defined in Jupyter notebook S1 (01_grid_agg.ipynb, section "2.2 Parameters"). By adjusting the parameter “GRID_SIZE_METERS = 100000” (cell 5, S1), the subsequent code and notebooks S2-S9 will react accordingly and produce figures and results at a different granularity. However, possible parameter settings have a lower limit of ~50 km, which is explained by effects of MAUP (see Andrade et al. 2020) and based on our initial data sampling with a Geohash of 5. It would also be relatively straightforward to adapt these notebooks to visualize higher resolution grids, using differently captured datasets for specific regions or areas (and not the world wide level). However, we did not test this.

de Andrade, S. C., Restrepo-Estrada, C., Nunes, L. H., Rodriguez, C. A. M., Estrella, J. C., Delbem, A. C. B., & Porto de Albuquerque, J. (2020). A multicriteria optimization framework for the definition of the spatial granularity of urban social media analytics. International Journal of Geographical Information Science, 00(00), 1–20. DOI: 10.1080/13658816.2020.1755039

> \\6. Figures are consistently unclear, or due to text size. They are hard or impossible to read.

Response: Thank you for pointing this out (reviewer 2 emphasized this too). We think there was an issue with the way PLOS processed our figures (png). While our original figures were high resolution (300 dpi), the PDF from PLOS looked very pixelated. We apologize for the inconvenience. We made sure that graphics submitted as part of this revision follow the PLOS guidelines for figures and also verified this with PACE (Analysis and Conversion Engine digital diagnostic tool, https://pacev2.apexcovantage.com/). We also increased font sizes in all figures, changed font consistently to Times New Roman, and reformatted figures selectively where we felt improvements were possible. In addition, figures and graphics in the data repository (S10, at this stage available at https://anonymous-peer12345.github.io/) are now available in the following formats:

• Interactive HTML (Bokeh), for zooming into maps, with additional information on hover. We used these interactive maps for examining results for the discussion in section §3.

• SVG (Vector graphics)

• PDF (Vector), for archiving purposes

• Rastered TIFF files in 600 dpi, for the PLOS one submission

• Rastered PNG files in 300 dpi, for web view

Note: The generated "Approval PDF" from PLOS still includes Figures that appear downsampled to low resolution. Since we have no influence over the PDF generation at PLOS, we would like to point out this issue. Full resolution figures (TIFF) are available through the URLs in the PDF, or from our data repository.

> Reviewer #2:

> The paper explores human global reactions to the sunset and the sunrise, using two datasets of geo-referenced photos collected from Instagram and Flickr, and aiming to understand the motivations behind taking and sharing sunset and sunrise photography. The authors attempted to analyze reactions across different groups, and in terms of aspects such as "what" is collectively valued "where", by "whom" and "when." To do this, the authors used relatively simple data analysis methods, specifically (a) using the HyperLogLog algorithm to count the number of distinct users sharing photos related to sunset or sunrise taken at different regions (i.e., different countries, or different cells in a global raster with a resolution of 4km) and at different months, (b) using TF-IDF heuristics to select discriminative terms associated to different regions, (c) using a spatial formulation of signed chi values to select regions that over and under represent the concepts of sunset and the sunrise.

> Overall, the paper is both sound and clearly written, presenting the results of what I consider to be an interesting analysis. Still, there are no methodological innovations being proposed (and I do have some questions regarding some of the methodological choices), and the topic addressed in the paper is relatively narrow and likely not particularly interesting to a broad audience.

Response: Thank you for these very supportive suggestions and helpful comments. It was indeed not our aim to improve individual methods, but rather combine and chain existing methods in a way that could serve as a robust "workflow template" for similar studies. We significantly revised the introduction to make our contributions to existing literature clearer and also added a paragraph that better explains why we explored a “relatively narrow” topic (p.3, ll. 28-41).

> I have some suggestions in terms of aspects that can perhaps be improved in the manuscript, which I list next.

> * The paper should perhaps further justify the choice of languages, besides english, that were considered for analysis with basis on a translation of the terms used for data collection (e.g., German, Dutch, and French). Although I do not consider this a requirement for acceptance of the paper, perhaps other popular languages could have been considered as well, including Spanish, Mandarin, Portuguese, Arabic, or Russian.

Response: Thank you for this question, we appreciate the opportunity to explain our choice of languages. Indeed, reviewer 1 also highlighted the importance to better explain and reflect on the consequences of this decision. We added a new paragraph to this effect (p. 10, ll. 200-209; see also changes in ll. 188-192). In particular, the choice of filter terms was primarily made based on the set of languages reasonably known to the authors of this paper, with the goal to avoid sampling errors from missed but important semantics in other languages. An example is given with our search terms for French (Tab. 1), where the language evolved to offer a more nuanced and richer set of meanings - one that is not available with the other three languages. The Coucher/Lever de Soleil, for example, is commonly used to reference the exact moment the sun sets or rises. In contrast, Coucher/Lever du Soleil typically refers to the overall experience of these passing events, (e.g.) with its colorful skies and light spectacles. Without knowing these cultural specificities, it is impossible to know which sampling biases are introduced by an inadequate list of (perhaps automatically translated) search terms. Nonetheless, Spanish, Mandarin, Portuguese, Arabic, or Russian are good examples for languages we would have wished to capture as well, since these appear to belong to countries where Flickr/Instagram use fluctuates more widely, based on a co-existence with other locally popular photo platforms. Please also note our answer to reviewer 1, where we pointed to other paragraphs that were important in the context of search terms and language selection.

> The authors can perhaps further justify the use of HyperLogLog, ideally also presenting a brief explanation on the paper. It is not entirely clear how/if the actual source data will be shared by the authors (the paper mentions a public repository, but at this time I could only access the pre-prepared maps and the Python notebooks. Also, I am not sure if sharing the source data is allowed by Flickr/Instagram), but I would argue that conducting the analysis with privacy-preserving methods would not be the main motivation in using HyperLogLog, although perhaps computational efficiency is an important motivation for this.

Response: Thank you very much for this question. We entirely understand the reviewers uncertainty regarding the HLL format and the data that is shared. Firstly, we added a paragraph (p.11, ll. 214-231) how HyperLogLog affects the collected data. Indeed, privacy was initially not our main motivation, but rather incidental. Given the immense volume, originally speaking, we were primarily interested in data minimization, as a means to practically perform the necessary number of iterations to tune parameters for different visualizations. However, data minimization provides benefits to both privacy and performance (see Dunkel et al. 2020). For instance, the dataset that abstracts all 302 Million IDs of geotagged posts from Flickr in our study is only 2.55 KB in size (see S9, cell 5). To our knowledge, this is the first applied study based on social media that makes use of HyperLogLog. The attached Jupyter Notebooks can serve as a base for transferring tools and methods to other studies. 

Regarding the actual data that is referenced as "data repository" (S10) in our study: It is correct that this repository with the actual notebooks (ipynb) and data (HLL) is not published yet. The referenced repository at https://anonymous-peer12345.github.io/ only contains HTML converts of notebooks and results. In the short timeframe of this revision, further changes were committed as part of the review process and it was not possible to finalize notebooks. We would like to take the time to make comments in notebooks consistent with the current version of the manuscript. It is also correct that we cannot share the raw data, only the abstracted HLL version. However, since we did not use raw data, only the captured HLL data in our analysis, this is sufficient to reproduce all results and figures in our study using the notebooks (S1-S9). Furthermore, we think that the shared expected frequencies (e.g. Flickr 300 million) can also be very useful to calculate chi in studies of other phenomena at global scales, for differently sampled data (e.g.) based on a different set of search terms. 

> The explanation associated to the use of TF-IDF should be improved. The authors mention the use of "spatial TF-IDF", but it is not clear what the "spatial" adaptation actually is, nor what is the difference towards standard TF-IDF. From the descriptions that are provided latter, I guess the authors are aggregating the textual descriptions from each photo and considering each of the spatial regions (countries or cells) as "individual documents", computing the TF and IDF components with basis on this aggregations (and hence they are able to use TF-IDF to get the most "discriminative" terms associated to each spatial region). However, this is not entirely clear, and should be better explained in the paper. The explanations associated to the TF-IDF equation should also be improved, given for instance that "f" is not a variable used in the equation (whereas "df" is).

Response: Thanks for this comment - you are correct and we have reformulated the text to make it clearer.

“To explore semantic patterns, we used two approaches. We ranked the terms for each country using term-frequency inverse document-frequency (TF-IDF) as a function of their global frequency (inverse document frequency) [51]. We define a ‘document’ as the set of all terms used by a single user per country. TF-IDF ranks terms used by many users in a country higher than those that are globally common, and ranked lists therefore reveal terms characterizing a grid cell or a country.” (p. 15, ll. 304-309)

We also reformulated the TF-IDF equation to avoid any ambiguity (p. 15), thank you very much for these suggestions.

> For comparing countries, the authors mention the use of "binary cosine similarity." This should also be further explained in the paper (e.g., does "binary" mean that the authors are considering vectors indicating only the presence of particular terms?) and, ideally, also further justified (why not use vectors of TF-IDF weights?).

Response: Again, thanks, you are quite right. We used the binary formulation - based on term occurrence for countries (instead of frequency vectors), because the results were better suited to compare similarities between vocabularies at a country level (as opposed to identifying prominent terms in individual countries). We did some tests using frequency vectors, but the results were noisy and still included many ‘filling’ words at the upper ranks, despite TF-IDF weights. We clarified this in the manuscript (p. 15, ll. 310-316).

> Many of the figures also seem to have a relatively low resolution and, for inclusion in a final manuscript, the authors should ideally provide vector versions of the images, instead of PNG files.

Response: Thank you - there appear to have been some issues with figure rendering, and this is dealt with in our response to reviewer 1 too. For convenience we copy our response to reviewer 1 here:

Response: Thank you for pointing this out (reviewer 2 emphasized this too). We think there was an issue with the way PLOS processed our figures (png). While our original figures were high resolution (300 dpi), the PDF from PLOS looked very pixelated. We apologize for the inconvenience. We made sure that graphics submitted as part of this revision follow the PLOS guidelines for figures and also verified this with PACE (Analysis and Conversion Engine digital diagnostic tool, https://pacev2.apexcovantage.com/). We also increased font sizes in all figures, changed font consistently to Times New Roman, and reformatted figures selectively where we felt improvements were possible. In addition, figures and graphics in the data repository (S10, at this stage available at https://anonymous-peer12345.github.io/) are now available in the following formats:

• Interactive HTML (Bokeh), for zooming into maps, with additional information on hover. We used these interactive maps for examining results for the discussion in section §3.

• SVG (Vector graphics)

• PDF (Vector), for archiving purposes

• Rastered TIFF files in 600 dpi, for the PLOS one submission

• Rastered PNG files in 300 dpi, for web view

Note: The generated "Approval PDF" from PLOS still includes Figures that appear downsampled to low resolution. Since we have no influence over the PDF generation at PLOS, we would like to point out this issue. Full resolution figures (TIFF) are available through the URLs in the PDF, or from our data repository.

---

## [Decision Letter · Decision Letter 1]

3 Jan 2023

From sunrise to sunset: Exploring landscape preference through global reactions to ephemeral events captured in georeferenced social media

PONE-D-22-16755R1

Dear Dr. Dunkel,

We’re pleased to inform you that your manuscript has been judged scientifically suitable for publication and will be formally accepted for publication once it meets all outstanding technical requirements.

Kind regards,

Jacinto Estima

Academic Editor

PLOS ONE

Additional Editor Comments (optional):

Most of the comments from the previous revision were addressed. Please do take into account the low resolution of images mentioned by reviewer 2.

Reviewers' comments:

Reviewer's Responses to Questions

**Comments to the Author**

1. If the authors have adequately addressed your comments raised in a previous round of review and you feel that this manuscript is now acceptable for publication, you may indicate that here to bypass the “Comments to the Author” section, enter your conflict of interest statement in the “Confidential to Editor” section, and submit your "Accept" recommendation.

Reviewer #1: All comments have been addressed

Reviewer #2: All comments have been addressed

2. Is the manuscript technically sound, and do the data support the conclusions?

Reviewer #1: Yes

Reviewer #2: Yes

3. Has the statistical analysis been performed appropriately and rigorously? 

Reviewer #1: Yes

Reviewer #2: Yes

4. Have the authors made all data underlying the findings in their manuscript fully available?

Reviewer #1: Yes

Reviewer #2: Yes

5. Is the manuscript presented in an intelligible fashion and written in standard English?

Reviewer #1: Yes

Reviewer #2: Yes

6. Review Comments to the Author

Reviewer #1: (No Response)

Reviewer #2: I have analyzed the revised version of the manuscript, and I was also one of the reviewers for the previous version (i.e., Reviewer 2).

The revised version of the manuscript has addressed most concerns put forward in the previous round of reviews, and the authors did a fine job in terms of further motivating the study and proposed approaches. There remains the fact that the paper is addressing a rather narrow topic, with no particular technical innovations, but the motivation is now clearer and I believe the manuscript can be accepted.

The suggestions for improvement that I had pointed before, e.g. regarding the explanations associated to TF-IDF and to the computation of cosine similarity), have been taken into account and, overall, I believe that the quality of the manuscript has improved.

The figures in the manuscript that was given to me for reviewing remain with some problems in terms of resolution, although the issue is likely due to the way PLOS processed the original files provided by the authors (and hopefully the issue can be corrected in the preparation of a camera ready version).

7. PLOS authors have the option to publish the peer review history of their article (what does this mean?). If published, this will include your full peer review and any attached files.

Reviewer #1: No

Reviewer #2: No
